# Removal of detritivore sea cucumbers from reefs increases coral disease

Cody S. Clements[1], Zoe A. Pratte[2], Frank J. Stewart[2] & Mark E. Hay [1]✉

Coral reefs are in global decline with coral diseases playing a significant role. This is especially true for Acroporid corals that represent ~25% of all Pacific coral species and generate much of the topographic complexity supporting reef biodiversity. Coral diseases are commonly sediment-associated and could be exacerbated by overharvest of sea cucumber detritivores that clean reef sediments and may suppress microbial pathogens as they feed. Here we show, via field manipulations in both French Polynesia and Palmyra Atoll, that historically overharvested sea cucumbers strongly suppress disease among corals in contact with benthic sediments. Sea cucumber removal increased tissue mortality of *Acropora pulchra* by ~370% and colony mortality by ~1500%. Additionally, farmerfish that kill *Acropora pulchra* bases to culture their algal gardens further suppress disease by separating corals from contact with the disease-causing sediment—functioning as mutualists rather than parasites despite killing coral bases. Historic overharvesting of sea cucumbers increases coral disease and threatens the persistence of tropical reefs. Enhancing sea cucumbers may enhance reef resilience by suppressing disease.

The large-scale removal of apex consumers is a hallmark of the Anthropocene and often cascades to impact the structure, function, and resilience of entire communities[1]. In some cases, these cascading impacts had decades-long delays between the actions initiating the cascades and their ultimate ecosystem-wide impacts. One example is the decrease in baleen whales due to increased harvest after World War II, causing predatory killer whales to switch to consuming a cascade of other large marine mammals, which 50 years later resulted in a dramatic decline in sea otters that allowed increases in herbivorous sea urchins and loss of kelp beds along much of the Alaskan coast[2]. If humans are lighting ecological fuses that burn for decades before their impacts are realized, these will be difficult to predict but critical to appreciate and counter.

Coral reefs are threatened by diverse ecological stressors associated with both contemporary and past human activities[3]. Coral cover has declined dramatically in recent decades, with disease being a major contributor[4–6], and with several coral diseases being associated with marine sediment[7–11]. Because sea cucumbers are detritivores that consume and process large amounts of sediment as they feed on

bacteria, microalgae, and organics[12–14], they might suppress sediment-associated pathogens and thus suppress coral diseases. However, sea cucumbers have been heavily harvested since the 1800s and are now functionally extirpated from many reefs[15,16]. World-wide harvests increased dramatically in the 1960s[17] and are continuing to increase; during the 2011–2020 period, annual wild harvests increased by ~30% and reached 57,700 tonnes of dried sea cucumbers[18]. This tonnage can be estimated to represent a harvest of more than 1,000,000,000 individuals/yr. Because recovery following harvest takes decades, or fails entirely, sea cucumber harvest is commonly a "mining of lootable resources" rather than a sustainable fishery[15,19], with harvests moving from shallower to deeper waters, from high value to lower value species, and into new geographic regions as previously fished regions are depleted[17,18]. Reliable estimates of historic sea cucumber densities are unavailable due to past large-scale harvesting, but densities of up to 50 individuals m$^{-2}$ were reported in some remote locations in the 1960s[20–22]. Their value as a food, ease of harvest, and inefficient reproduction at low densities led to the widespread collapse of populations[16], even with management intervention[23]. Because sea

[1]School of Biological Sciences and Center for Microbial Dynamics and Infection, Georgia Institute of Technology, Atlanta, GA, USA. [2]Department of Microbiology and Cell Biology, Montana State University, Bozeman, MT, USA. ✉e-mail: mark.hay@biology.gatech.edu

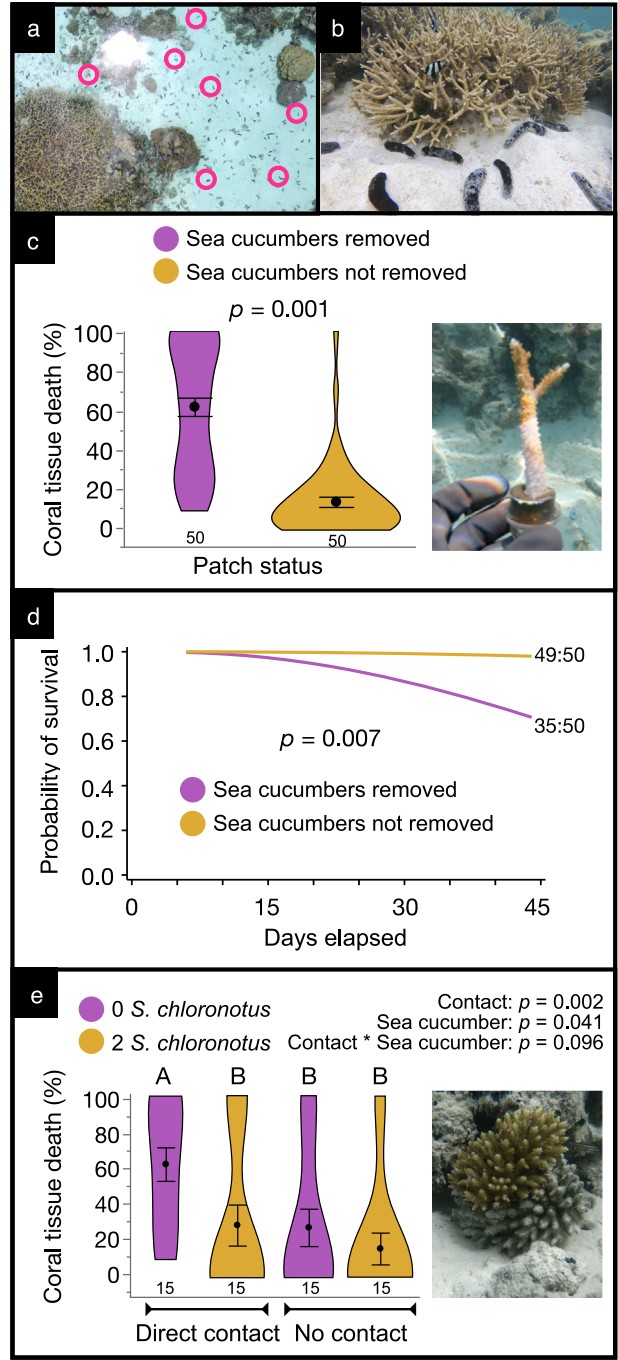

**Fig. 1 | Effects of sea cucumber removal on coral tissue death and mortality.** **a** Aerial view of a sand patch in the field with feeding sea cucumbers (*Holothuria atra*, pink circles denote example individuals) and a thicket of *Acropora pulchra* coral (bottom left). **b** A typical sand patch in the field, with *H. atra* and *A. pulchra* coral with turf algae at the base of the coral thicket. **c** *A. pulchra* percent tissue death (mean ± SE) with an inset picture of an *A. pulchra* outplant experiencing the typical pattern of tissue mortality from the base up. Total numbers of coral outplants assessed per treatment are indicated below each violin plot ($n = 10$ sand patches, each holding 5 coral outplants for a total of 50 outplants per treatment). *P* value derived from a one-way, permutation-based linear mixed-effects (LME) model. **d** *A. pulchra* probability of survival through time as a function of sea cucumber presence or removal. Ratio of surviving coral outplants at 45 days to total number of outplants are indicated to the right of survival curves ($n = 10$ sand patches, each holding 5 coral outplants for a total of 50 outplants per treatment). Survival curves and *p* value derived from two-tailed, random-effects Weibull regression. **e** *Acropora nasuta* percent tissue death (mean ± SE) as a function of sea cucumber (*Stichopus chloronotus*) presence or absence and outplant contact with sediment. Total numbers of coral outplants assessed per treatment are indicated below each violin plot ($n = 15$ outplants per treatment). *P* values derived from a two-way, permutation-based linear mixed-effects (LME) model. Letters indicate significant groupings via a post hoc permutation test for multiple comparisons. The inset image is of an *A. nasuta* colony exhibiting partial mortality where in contact with the sediment. Source data are provided as a Source Data file.

parasites that kill coral bases to grow the algal gardens from which they feed, but it is possible that in the larger community context, they function as mutualists, protecting the coral from the dangers of disease in the sediment below the coral.

Here we show that two *Acropora* species associated with sandy environments on both a high island with a human population and a low island (atoll) with no human population experienced lower frequencies of disease and greater survivorship when sea cucumbers were present than when they were removed. Incidents of coral disease consistently entailed contact with benthic sediments, but sea cucumber presence both remediated these adverse effects and altered sediment microbiomes, suggesting that their feeding may help suppress sediment-associated pathogens. This may be especially important for reef-building *Acropora* species that commonly spread via fragmentation[28,29]. Furthermore, we show that once corals grow to sizes that attract farmerfishes that cultivate gardens of filamentous turf algae on coral bases, the turf algae separating live coral tissue from contact with sediment strongly protects corals from infection. Thus, sea cucumbers and farmerfish turfs may act as ecological defenses against infection from marine sediments that are suspected reservoirs of coral disease[7-11].

## Results and discussion

Using reefs in Moʻorea, French Polynesia, and Palmyra Atoll where some areas still support sea cucumbers at densities of several/m² (see Fig. 1 a, b)[30], we experimentally evaluated the effects of sea cucumber removal on the frequency and extent of disease damage to co-occurring corals. When individuals of the sea cucumber *Holothuria atra* were removed from ($n = 10$ patches) or left at natural densities ($n = 10$ patches) in interspersed sand patches (ranging from 6.3 to 12.2 m² each) in lagoonal areas of Moʻorea for 45 days, tissue mortality for five *Acropora pulchra* corals planted into each patch with their bases contacting sediment was on average 370% greater in sea cucumber removal ($62.2 ± 4.7$; mean ± SE) vs. control ($13.3\% ± 2.6\%$) patches ($n = 50$ per treatment, Permutation-based LME, $F = 34.219$, $df = 18$, $p = 0.001$; Fig. 1c). Whole colony mortality was 1500% greater (30% vs 2%; Survival Analysis, Hazard ratio $0.056 ± 0.060$, $p = 0.007$, Fig. 1d). Thus, sea cucumber presence within patches reduced the risk of whole colony mortality by ~94%.

Moʻorea is a high island with a human population of 130 km⁻² and so might have more runoff and pollution supporting sediment-associated microbes than low islands without such input. However,

cucumbers clean sediment by consuming bacteria and organics[12], sea cucumber removal could lead to outbreaks of microbial pathogens or otherwise alter sediment microbial processes in threatened ecosystems such as coral reefs, especially as humans are heating the ocean and adding organics, both of which increase microbial metabolism and the risk of upregulating pathogenicity[24-26].

An additional consideration is that coral diseases associated with sediment exposure often seem to occur following direct contact with sediment or turbulence that moves sediment onto corals[7-11]. If direct contact risks infection, then species, like some *Acropora*, that form thickets directly on sand should be at especially high risk. However, once thickets are established, their bases are commonly infested with damselfishes that kill coral bases, cultivate algal gardens on these bases[27], and thus separate live coral tissues from sediment contact via a barrier of filamentous turf algae. These farmerfishes appear to be

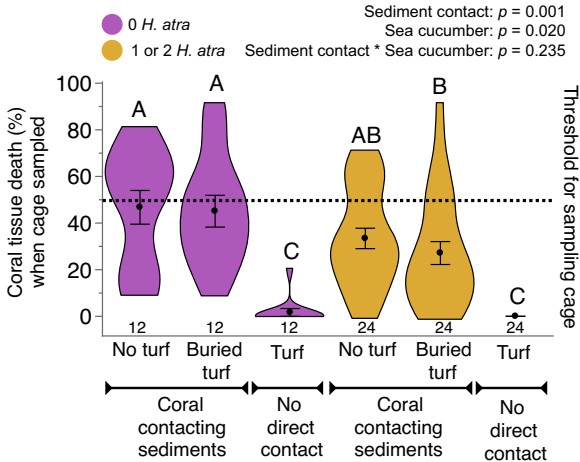

**Fig. 2 | *Acropora pulchra* percent tissue death (mean ± SE) as a function of presence or absence of sea cucumbers and coral outplant type.** Outplants with no turf and buried turf were planted in such a way that living coral tissue directly contacted the sediment, while turf samples reflect living coral tissue that was separated from the sediment by farmerfish turf algae on the base of the outplant. Total numbers of corals assessed per treatment are indicated below each violin plot. *P* values derived from a two-way, permutation-based linear mixed-effects (LME) model (*n* = 15 per treatment). Letters indicate significant groupings via a post hoc permutation test for multiple comparisons. Source data are provided as a Source Data file.

when we conducted a similar experiment assessing effects of sea cucumbers on coral health on Palmyra Atoll, which has minimal land area (3.9 km$^2$), all at low elevation, and no permanent human population, we documented similar effects. In the lagoon at Palmyra Atoll, when outplants of the coral *Acropora nasuta*, either in contact with or elevated 2–3 cm above benthic sediment, were enclosed in 50 × 50 cm square cages with or without two *Stichopus chloronotus* sea cucumbers (*n* = 15 cages per treatment) for 14 days, both contact with the sediment (Permutation-based LME, *F* = 14.239, *df* = 28, *p* = 0.002) and sea cucumber exclusion (Permutation-based LME, *F* = 3.235, *df* = 28, *p* = 0.041) increased coral tissue mortality (Fig. 1e). Corals without sea cucumbers and in contact with sediment experienced ~63 ± 10% tissue mortality while those without sea cucumbers but elevated above the sediment experienced only 27 ± 11% mortality. Presence of sea cucumbers negated this difference; in the presence of cucumbers, tissue mortality of corals in contact with sediment did not differ significantly from mortality of elevated corals (~28 ± 12% versus 15 ± 9%, respectively; Fig. 1e). Thus, despite evaluating different sea cucumbers, different corals, a high island populated by humans and a low island with minimal human impact, and different sites separated by ~3000 km, sediment-feeding sea cucumbers suppressed coral tissue death associated with sediment contact in both locations. *A. pulchra* commonly grows directly on sediment (Fig. 1a) while *A. nasuta* attaches to hard substrates, but these are often low and covered by, or surrounded by, sediment that contacts coral bases (inset image of Fig. 1e).

In all cases of disease, corals developed a white band of dead tissue at the sediment-coral juncture, with this band progressing up the coral branch over time (inset image of Fig. 1c), similar to white band disease[31]. Others have reported coral disease associated with sediment[7–11], and outplants of *Acropora* species in the Caribbean commonly die with a white band progressing along the coral branch, often from the base up[5,32] like we observed in our Pacific experiments. Additionally, sea cucumber presence was recently shown to enhance the protective potency of polar extracts from *Acropora cytherea* against the coral pathogen *Vibrio coralliilyticus*[30], suggesting that sea cucumbers may enhance coral defenses against pathogens as well as

suppress sediment-associated microbes. However, sea cucumber suppression of coral disease and mortality under field conditions had not previously been assessed or demonstrated.

Despite Acroporid corals sickening and dying when in contact with sediment in the absence of sea cucumbers, these corals, as well as others, growing on or near sediment are thought to commonly spread via dispersal of fragments across reef sediment or rubble[31,32], and there are large thickets of healthy *A. pulchra* growing on sediment at our study site in Mo'orea and in several areas without sea cucumbers. However, these healthy *Acropora* thickets all host *Stegastes* farmerfishes that culture and defend gardens of filamentous algae on the coral bases, thus separating live coral tissues from direct contact with sediment. These farmerfishes protect *A. pulchra* from coral eating fishes[27,33], but their algal gardens also might protect corals from disease by providing a barrier against sediment-associated pathogens.

To test this hypothesis and how the protective function varied as a function of sea cucumber density, we erected 50 × 50 cm cages on sandy areas of the Mo'orea lagoon, placed zero, one, or two *H. atra* in each cage (*n* = 12 cages/treatment), interspersed treatments, and let benthic communities develop for seven days. On day 7, we collected sediment from each cage to quantify the effects of sea cucumbers on sediment microbiomes and into all cages, we placed three treatments of *A. pulchra*: (i) a coral branch with turf algae on the base separating live coral tissue from sediment by ~2.5 cm of turf (hereafter referred to as turf), (ii) a coral with equivalent turf algae on its base but with the turf buried so that sediment contacted the coral's live tissue at the turf-coral juncture (hereafter buried turf), or (iii) a coral lacking turf on its base and its live tissue in direct contact with benthic sediment (hereafter no turf). Coral branches for the experiment were collected from numerous *A. pulchra* patches in the area near our experimental cages, and this experiment was monitored daily for 36 days, visually estimating % tissue death of each coral each day.

To evaluate microbes associated with the white band-type symptoms, when the band of dead tissue on any coral in a cage was estimated to be ≥50% of the coral branch's height, all corals in that cage were collected, and diseased tissue was sampled at the juncture of the white band and healthy tissue and at the healthy tissue about 1 cm from the coral tip. Other corals in that cage were sampled 1 cm from the tip (hereafter distal samples) and at the lowest portion of live coral tissue (hereafter basal samples) so that microbiomes of corals contacting sediment (100% were diseased) versus separate from sediment by turf algae (only 3% were diseased) from the same cage and time could be compared via Illumina sequencing. Because there is spatial variance in timing of disease onset, this sampling scheme allowed i) all treatments at a small-scale site (the cage replicate, which served as a block for the three outplant types) to be sampled synchronously and minimize effects of spatial as opposed to treatment variance and ii) assessing the coral microbiome for possible identification of the disease-causing organism at the site of the moving white-band of disease. Had we sampled all replicates at a uniform time, diseased treatments in some replicates would have progressed to complete mortality while no outplants in others would have been infected – making sampling at the disease front impossible for many replicates. When we sampled corals within any cage, we also collected sediment for microbiome analyses from that cage. All microbiomes were evaluated with sequences grouped into both exact sequence variants (ESVs) and into clusters sharing >90% similarity (approximately the genus level), thereby allowing both fine and more general levels of taxonomic resolution.

Disease impact as a function of coral treatment did not differ between cages with one versus two sea cucumbers (Supplementary Fig. 1, Permutation-based LME post hoc test, *p* = 0.856), so we pooled these treatments and compared treatments with versus without sea cucumbers. Both sea cucumber presence and the presence of turf preventing direct contact of coral tissue with sediment significantly decreased coral tissue loss to disease (Fig. 2; Permutation-based LME,

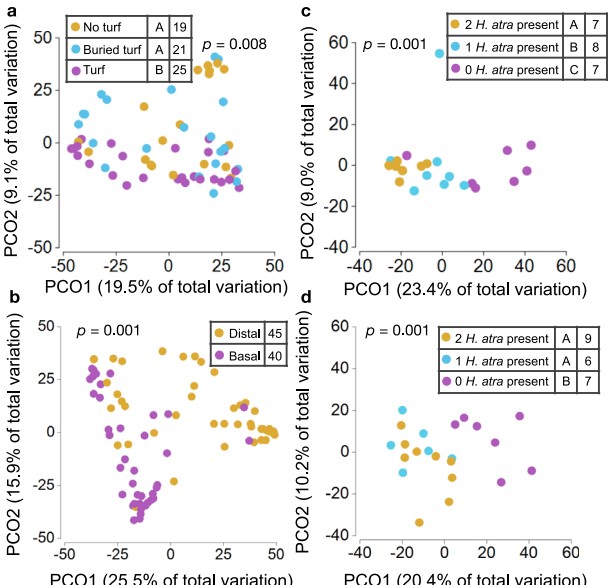

**Fig. 3 | Effects of coral (*Acropora pulchra*) outplant type, sea cucumber (*Holothuria atra*) density, and basal versus distal samples on coral and sediment microbiomes. a** PCoA (Bray-Curtis dissimilarity index) and one-way PERMANOVA analysis of microbiome composition (beta diversity) of basal coral samples as a function of outplant type. Outplants with no turf and buried turf were planted so that living coral tissue directly contacted the sediment, while turf treatments had coral tissue separated from the sediment by basal turf algae. Letters within legend indicate significant groupings via a post hoc permutation test for multiple comparisons. Total numbers of samples assessed per treatment are indicated to the right of the letter report. **b** PCoA (Bray-Curtis dissimilarity index) and one-way PERMANOVA analysis of microbiome composition (beta diversity) of basal and distal samples from coral outplants within experimental enclosures. Basal samples derived from living tissue immediately adjacent to dead or dying tissue at the outplant's base and distal samples from living tissue -1 cm from the branch tip of the outplant. The total numbers of samples assessed per treatment are indicated within the legend. **c, d** PCoA (Bray-Curtis dissimilarity index) and one-way PERMANOVA analysis of microbiome composition (beta diversity) from surficial sediment within cages containing two, one, or zero sea cucumbers after **c** seven days of sea cucumber feeding and **d** immediately prior to sampling corals within cages (ranging from 7 to 36 days of coral exposure). Letters within legends indicate significant groupings via post hoc permutation tests for multiple comparisons. Total numbers of samples assessed per treatment are indicated to the right of the letter reports. Source data are provided as a Source Data file.

sediment contact: $F = 40.578$, $df = 68$ $p = 0.001$, sea cucumber presence: $F = 7.930$, $df = 34$, $p = 0.020$, interaction: $F = 1.541$, $df = 68$, $p = 0.235$). For corals contacting sediment, tissue mortality rates were reduced 26–42% by the presence of sea cucumbers and by 94–100% when turf algae prevented live coral tissue from contacting sediment.

Microbiomes from distal portions of corals (all appeared healthy) did not differ in composition among treatments (Supplementary Fig. 2; PERMANOVA, pseudo $F = 1.117$, $df = 2$, $p = 0.299$), indicating that disease effects were localized near the white band of dead tissue and justifying a focus on basal samples. Basal samples from corals in direct contact with sediment (100% were diseased) differed significantly in microbiome taxonomic composition from basal samples of corals separated from sediment by turf (only 3% were diseased; Fig. 3a; PERMANOVA, pseudo $F = 1.833$, $df = 2$, $p = 0.008$); additionally, the basal samples of infected corals (no turf and buried turf treatments) differed significantly from distal samples in those treatments (PERMANOVA, pseudo $F = 10.447$, $df = 1$, $p = 0.001$, Fig. 3b). Pairwise comparisons of basal microbiomes did not differ between corals with turf embedded in the sediment and those lacking turf (Fig. 3a; PERMANOVA Pair-wise test, $p = 0.413$,), so we pooled treatments contacting sediment and compared them to the treatment separated from

sediment contact. Forty-four ESVs were significantly more abundant in basal samples contacting sediment, while only 5 ESVs were more abundant in basal samples protected by a band of turf algae (Supplementary Data 1). For basal samples of corals contacting sediment, the differentially abundant ESVs exhibiting relative abundances >1% included: Arcobacteraceae (unknown genus, 2.84%; genus *Halarcobacter*, 1.99%), Flavobacteriaceae (genus *Aquibacter*, 1.83%, 1.25%), Pseudoalteromonadaceae (genus *Algicola*, 2.33%), Nitrincolaceae (genus *Neptuniibacter*, 1.56%), Rhodobacteraceae (unknown genus, 1.57%), Francisellaceae (genus *Caedibacter*, 1.08%), and Alteromonadaceae (genus *Agaribacter*, 1.03%). These groups are commonly associated with coral stress and disease[7,34–36], including diseases hypothesized to derive from marine sediment (e.g., Stony Coral Tissue Loss [SCTL] and white band syndromes)[7–11]. Thus, sediment contact facilitates disease, and both sea cucumbers and farmerfish turfs protect corals from sediment-associated pathogens.

Sediment microbiomes differed in community composition as a function of sea cucumber presence or absence (Fig. 3c, d), but at neither the ESV-level nor the 90% similarity level were the microbes that increased following sea cucumber removal those that were common in diseased basal samples of corals (contrast Supplementary Data 1, 2 and 3e & 4). The microbes enriched in diseased coral tissue might be causing the disease or might be opportunists responding to necrotic tissue caused by other species or processes[35,37]. Rigorous identification of pathogens infecting corals and other marine invertebrates has proven difficult despite decades of effort[38,39] and remains so despite some recent successes[40,41].

The lack of a clear spike of a suspected pathogen may not be surprising if the causative microorganism(s) is a natural component of the environmental or host microbiome and only becomes pathogenic if circumstances allow. This is the nature of opportunistic pathogens, for example, various *Enterococcus* spp. that are commensal in the human GI tract but disease-causing if disturbance (e.g., wounding, ulcers, surgeries) allows spread to other body sites[42]. Such scenarios could also involve communities of microorganisms (a polymicrobial consortium), analogous to what has been suggested in black band disease, with host damage driven by diverse metabolic outcomes associated with microbial overgrowth[43]. Disease-causing microorganisms may also be hard to identify if they occur at low frequency but exert a pathogenic effect indirectly by remodeling a commensal microbiome into a harmful one, for example through changes in cell number, relative abundance, or metabolic activity[44]. These so-called keystone pathogens are implicated in human consortia diseases (e.g., periodontitis) and have been hypothesized as agents of coral disease[39]. Additional causes could include ciliates[40,41], fungi[45], or chemical changes at the sediment-coral juncture as microbes or organics accumulate in the sediment[46] without being removed or bioturbated by sea cucumbers.

Marine sediment, on which sea cucumbers feed, may be reservoirs for coral diseases[7–11,47] and sea cucumbers can process as much as 80 kg of sediment individual[−1] year[−1] as they feed on bacteria, microalgae, and other sediment-associated organic material[13]. Cyanobacterial or microalgal mats can form following sea cucumber removal[13,30] and can coincide with reductions in coral defenses against pathogens[30]. Thus, historic sea cucumber harvesting may have suppressed waste management on reefs at a time when disease and other stressors are increasing and decimating corals[5–7], the foundation species of these ecosystems. The paucity of sea cucumbers on modern reefs, overharvest of remaining populations, and lack of documentation regarding historic natural densities before wide-scale harvesting make assessing effects of historic, natural densities of sea cucumbers challenging; however, less commercially desirable sea cucumbers have avoided overexploitation in a few remote or protected locations and remain amenable to experimental manipulations, like those conducted here. As an example of historic patterns, sea cucumbers in French

Polynesia were harvested for export as early as 1810[48]. Quantities exported were first reported in the 1930s (60,000 kg), but were only documented sporadically until 2008, when export to markets in Asia led to rapid overharvest of most remaining populations[49]. Exports of dried sea cucumber peaked at 126 tonnes in 2011 and ultimately resulted in a nationwide moratorium on sea cucumber harvest in 2012. While many commercially valuable species were completely depleted, less-desired species such as *Holothuria atra* remain in some areas[50] and can still be found at mean densities of $7.1 \pm 4.7/m^2$ (mean ± SE), with densities ranging from 1.7 to $23.0/m^2$ across the 30 patches that were assessed (Fig. 1a, b)[30]. Preserving, or cultivating and outplanting, sea cucumbers in areas undergoing coral restoration or areas subject to disease outbreaks could potentially suppress coral losses to disease. The primary sea cucumber species used here (*H. atra*) is of little commercial value and thus at low risk of being harvested if populations are protected or enhanced.

For corals like *A. pulchra* (or *A. cervicornis* in the Caribbean) that can form thickets on sandy sediment and commonly spread via fragmentation, our outplants in contact with sediments would mirror the rigors of fragmentation and vegetative spread. Our data suggest that sea cucumbers can provide critical benefits by preventing death due to disease during this colonization stage. Once colonizers persist and grow to a size sufficient to offer shelter and substrate to farmerfishes, the advantages of sea cucumbers appear less critical. The beneficial effect of farmerfish gardens in separating live coral tissues from direct contact with sediment allows continued growth and spread of the patch even without sea cucumbers nearby.

Loss of detritivores and scavengers such as vultures can alter nutrient cycling and energy flow[51], facilitate mesopredator increases that affect trophic cascades[52], and has been implicated in the spread of disease among wildlife and to humans[53,54]. The ecological consequences of the long-term and global-scale removal of major detritivores deserves more consideration. Here, we show that decades to centuries of sea cucumber harvest may contribute to modern outbreaks of coral disease due to removal of these sediment cleaners and consumers of microorganisms, potentially including pathogens. Today's explosive growth of coral diseases may be caused, at least in part, due to long-burning, ecological fuses lit in the 1800s by massive harvests of sea cucumbers. Restoring the essential cleaning services that these detritivores provide may be especially critical to modern oceans due to ocean warming and organic enrichment – both of which enhance microbial growth, metabolism, and pathogenicity[24,26].

## Methods
### Study areas and natural sand patch experiments
We assessed the effects of the sea cucumber *Holothuria atra* on the coral *Acropora pulchra* within a shallow fringing reef on the north coast of Mo'orea, French Polynesia (17.4894° S, 149.8825° W). *H. atra* dominated the sea cucumber assemblage of our study reef, occurring at a mean density of $7.1 \pm 4.7/m^2$ (mean ± SE), with densities ranging from 1.7 to $23.0/m^2$ across the 30 sandy patches that were assessed[26]. We chose 20 of these patches (study area = ~8900 $m^2$, patch areas ranging from 6.27 to 12.15 $m^2$), randomly assigned each to a removal or control plot, and either consistently removed (every 1–2 days) or did not remove sea cucumbers to test how sea cucumber removal impacted *A. pulchra* tissue mortality and survivorship. Sand patches were bordered by bommies of dead coral or living colonies of *A. pulchra*, *Porites lobata*, *Porites rus*, *Pavona cactus*, *Pocillopora damicornis*, *Montipora* spp., etc. that were common throughout the study area (e.g., see Fig. 1a).

At the initiation of the experiment, sea cucumbers were removed daily (removals) or left in place (controls) for seven days to condition sand patches for subsequent coral planting, after which five *A. pulchra* outplants approximately 8–10 cm in length were embedded in the sediment of each patch, with % coral tissue mortality and outplant survival monitored at approximately 2-day intervals for 45 days (50 corals treatment⁻¹, 100 corals total). The corals were initially fragmented from numerous *A. pulchra* thickets adjacent to our study area, individually embedded within the cutoff necks of inverted plastic bottles using Z-Spar Splash Zone epoxy (see[55]) and screwed into upturned bottle caps attached to ~$7 \times 7$ cm pieces of metal gridded mesh that could be slid into the sediment to hold each coral upright. To prevent feeding by coral consumers, each coral was caged within 1 $cm^2$ metal screening. Corals were embedded within their sand patches so that living basal coral tissue was in direct contact with the sediment as would occur following natural fragmentation. Every other day for 45 days, we counted sea cucumbers, maintained removal treatments, cleaned cages, and quantified *A. pulchra* tissue mortality in each patch.

We used a permutation-based, linear mixed-effects (LME) model in the R[56] package predictmeans[57] to compare differences in percent tissue mortality between corals in plots where sea cucumbers were removed versus not removed. Patch type (removed vs. not removed) was treated as a fixed factor, with replicate patches treated as a random effect nested within patch type. To compare overall coral survivorship at 45 days, survival curves and analyses were generated using random-effects Weibull regression in Stata (version 17), with patch type as a fixed effect and replicate patches treated as a random effect nested within patch type.

### Sea cucumber enclosure experiments
In the lagoon on Palmyra Atoll (5°52'42.6"N 162°04'09.8"W), we conducted an experiment with a similar goal of determining the effects of a common sea cucumber on coral health but enclosed or excluded the locally abundant sea cucumber *Stichopus chloronotus* and assessed impact on the coral *Acropora nasuta*. We did not perform large-scale removals (as we did in Mo'orea) due to Palmyra Atoll being a reserve managed by the US Fish and Wildlife Service and their permitting process limits such large-scale manipulations when more modest manipulations (in this case smaller cages) can address the question posed. At this site, *S. chloronotus* occurred at mean densities of $2.7 \pm 0.7 m^{-2}$ ($n = 30$ haphazardly located $m^2$ quadrats), but were clumped on patches of low-lying, often sediment-covered, hard substrate, with these areas reaching densities of 10–13 *S. chloronotus* $m^{-2}$. *A. nasuta* occurred occasionally on these low-lying, sediment-covered, hard substrates. Within an area of about 160 $m^2$, we erected thirty $50 \times 50 \times 12$ cm cages constructed of plastic 1 $cm^2$ mesh, planted two *A. nasuta* either in contact with or elevated 2–3 cm above benthic sediment into each cage (planted as described above), and randomly assigned cages to contain either zero or two *S. chloronotus* sea cucumbers ($n = 15$ for each treatment combination). Coral portions for the experiment were collected from 15 *A. nasuta* colonies in the East Lagoon of Palmyra Atoll where our experimental cages were located. Cages and coral tissue death were monitored daily for 14 days, with data from the final day used to contrast effects of sea cucumber presence and sediment contact on coral condition. We used a permutation-based, linear mixed-effects (LME) model in the R[56] package predictmeans[57] to compare differences in percent coral tissue mortality as a function of sea cucumber presence vs. absence, as well as coral direct contact vs. no contact with sediment. Sea cucumber status and coral-sediment contact status were treated as fixed effects and cage treated as a random effect. Subsequent comparisons were conducted using a post hoc permutation test for multiple comparisons using predictmeans[57].

In Mo'orea, to assess the impact of sea cucumber removal on sediment- and coral-associated microbiomes, as well as how farmerfish turf on the base of corals might affect disease prevalence, we erected thirty-six $50$ cm × $50$ cm × $12$ cm tall cages using 1 $cm^2$ grid metal screening to contain or exclude sea cucumbers and prevent access by

coral consumers. Cages were situated in an ~85 m² sand patch within the fringing reef area utilized in our initial experiment described above and were separated from adjacent cages by ≥60 cm, creating a 6 × 6 grid of enclosures. Each cage was stocked with either zero, one, or two *H. atra* (12 cages treatment⁻¹) that were approximately 9–14 cm in length, as is typical for individuals at our site. Density treatments were assigned at random. Cages were inspected daily to ensure that density treatments were maintained (they were), and sea cucumbers outside of cages were removed daily from within about 10 m of the 2 m deep area where cages were situated. All cages were brushed every other day to prevent fouling.

Seven days after applying sea cucumber treatments, sediment samples were taken for microbiome analyses by scraping 30–40 mL of surficial sediment from the top ~5 mm of each caged area into a small Whirl-Pak. Samples were immediately placed on ice and stored in a −80 °C freezer upon return to shore. Following sediment sampling, three *A. pulchra* outplants were embedded into the sediment of each cage (108 outplants total) to test the potential effects of (i) sea cucumber density, and (ii) protective effects of farmerfish-cultivated turf algae on coral health and microbiomes (see below). Corals used were approximately 8–10 cm in length and were initially fragmented from numerous *A. pulchra* thickets adjacent to our study area and outplanted using the methods described above.

Of the three outplants included in each cage, two were fragmented from colonies in the field in such a way as to include farmerfish-generated turf algae at their base, while the third was fragmented so that it lacked turf at its base. These three outplants were embedded into the sediment as follows: (i) coral lacking turf planted in direct contact with benthic sediment (hereafter "no turf"), (ii) coral separated from direct contact with sediment by turf algae growing at its base (hereafter "turf"), or (iii) coral with turf on its base, but embedded more deeply into the sediment so that the living coral tissue was in direct contact with the sediment (hereafter "embedded turf"). Percent coral tissue mortality among outplants was visually estimated daily for 36 days. The microbiomes of all corals and the sediment within a cage were sampled when one or more outplants within that cage exhibited ≥50% tissue mortality or when the experiment was terminated on day 36. This sampling scheme was used due to preliminary assays indicating considerable location specific variance in the rate at which corals sickened when in contact with sediment. Because each cage contained a block of all coral transplant types, this allowed us to minimize the variance due to location and focus on the response of coral outplant type. It also allowed sampling microbiomes associated with coral disease because we could sample diseased sections of the first infected coral (when the white-band type disease had progressed 50% of the length of the coral) before the disease killed the entire outplant, eliminating the evidence of microbes potentially causing the disease. If we had let the entire experiment run to some randomly chosen time, some replicates may have been dead and some uninfected (both situations would have prevented sampling of diseased areas).

To determine the microbes associated with the coral disease that spread upward from the base of all infected outplants, sampling involved taking ~0.5 g clippings from: (i) healthy, living tissue ~1 cm from the branch tip (hereafter "distal"), and (ii) living basal tissue (hereafter "basal"). Basal tissue of outplants with turf separating live coral tissue from the sediment surface was sampled immediately adjacent to turf algae, or immediately adjacent to dying tissue (for the single replicate that experienced any tissue mortality in this treatment). For corals with no turf or embedded turf, basal tissue was sampled immediately adjacent to the junction of dead or dying tissue; 100% of these outplants exhibited some level of tissue mortality at the base of the outplant. All samples were placed in 2 ml cryovials with RNA/DNA stabilizing buffer (25 mM sodium citrate, 10 mM EDTA and 70 g ammonium sulfate per 100 ml solution, pH 5.2), immediately

placed on ice, and stored in a −80 °C freezer upon return to shore. During sampling of outplants within each enclosure, we also resampled benthic sediment in the same manner described above, but sampling varied in timing due to corals in the different replicates achieving the ≥50% dead tissue status at different rates. We used Fisher's exact tests to assess differences in the frequency of mortality-initiated sampling among sea cucumber treatments (i.e., zero, one, or two *H. atra*) and among outplant types (i.e., no turf, turf, or embedded turf). To compare differences in percent tissue mortality when corals were sampled for microbiome analyses, we used a permutation-based, linear mixed-effects (LME) model in the R[56] package predictmeans[57]. Sea cucumber status and coral outplant type were treated as fixed effects and cage as a random effect. Subsequent comparisons were conducted using a post hoc permutation test for multiple comparisons using predictmeans[57].

### DNA extractions and sequencing of the 16 S ribosomal RNA gene
DNA was extracted from each sediment or coral sample by placing approximately 250 mg of sediment (n = 46) or a small (roughly 5 mm³) fragment of coral (n = 142) that included skeleton, tissue, and mucus into the bead-beating tube of the Qiagen PowerSoil DNA extraction kit. DNA was then extracted according to the manufacturer's protocol. Extraction blanks were performed for each kit utilized (n = 5) and carried throughout the amplification and sequencing process. The V4 region of the 16 S rRNA gene was amplified for each sample using the primers F515 (Parada) (5′-GTGYCAGCMGCCGCGGTAA-3′) and R806[58] (5′-GGACTACNVGGGTWTCTAAT-3′), each of which were appended with unique barcode identifiers for dual indexing[59]. PCR conditions were as in Caughman et al.[60]. Each sample was amplified in duplicate to produce technical replicates, with each technical replicate having its own unique barcode identifiers. Amplicons from each sample were then randomly pooled in equimolar concentration into two pools and sent to Georgia Genomics and Bioinformatics Core for further processing and sequencing. To remove large primer dimers that formed during the amplification process, each pool was heat-denatured and then size selected using magnetic beads targeting amplicon fragments of 300–350 bp. Each pool was then sequenced using Illumina MiSeq technology and a paired-end 500 cycle kit with V2 chemistry, with 10% PhiX to increase read diversity.

Raw.fastq files were imported into QIIME2 version 2021.11[61] and denoised, merged, and checked for chimeras using DADA2[62] with the parameters –p-trim-left-f 25 –p-trim-left-r 25 –p-trunc-len-f 175 –p-trunc-len-r 175. Taxonomy was assigned to each Exact Sequence Variant (ESV) produced by DADA2 using the Silva 138 database, and all chloroplast and mitochondria ESVs were removed from the dataset. Beta-diversity plots were generated in QIIME2 to visually assess the quality of technical replicates. Those samples clustering with the extraction blanks were removed, along with technical replicates that did not cluster with one another, indicating stochastic amplification. This quality control resulted in the removal of all blanks, 13 coral samples, and two sediment samples. Technical replicates of those samples passing quality control were then merged, resulting in a final dataset comprising 129 coral samples and 44 sediment samples. Samples were then rarefied to 1624 reads, with the rarefied table used for all subsequent analyses, with the exception of the ESV differential abundance analyses.

### Microbiome data analyses
Analyses of microbiome data were conducted in a stepwise manner to test whether and how microbiomes differed among *A. pulchra* outplants and associated sediment. Questions, in order of analysis, were:

i. *Among distal samples from coral outplants, did microbiomes differ as a function of outplant type or sea cucumber density?*
ii. *Among basal samples from coral outplants, did microbiomes differ as a function of outplant type or sea cucumber density?*

iii. *Among outplants in direct contact with sediment (i.e. no turf and embedded turf), did microbiomes differ as a function of sampling location or sea cucumber density?*

iv. *Did sediment microbiomes differ as a function of sea cucumber density?*

In each case, Bray-Curtis dissimilarity values were calculated using the distance function in PRIMER-e[63]. Principal coordinate analysis (PCO) and corresponding tests for differences in microbiome composition (permutational multivariate analysis of variance, PERMANOVA) were implemented in PRIMER-e via tests with one (Q i-ii) or two (Q iii-iv) factors. If significant, subsequent comparisons were conducted using the "Pair-wise test" option within the PERMANOVA function of PRIMER-e. To test for differences in microbiome variability (dispersion, measured as deviation from the centroid), we used the PERMDISP function in PRIMER-e for all relevant analyses, including subsequent pairwise comparisons when significant. Alpha diversity (ESV richness, Shannon diversity) of relevant datasets was calculated using the core-metrics-phylogenetic function in QIIME2. Relevant comparisons were conducted using permutational ANOVA in the R[56] package predictmeans[57]. To identify differential ESV abundances, non-rarefied tables were imported into R through the packages qiime2R and phyloseq. In cases where pairwise comparisons were statistically indistinguishable in broader-scale microbiome metrics (e.g., microbiome composition), these non-significant sample groupings were collapsed into a single grouping, and ESV abundances were compared to those of groupings that differed significantly in broader-scale microbiome metrics. DESeq2[64] was then used to detect differential ESV abundance among relevant sample groupings, with a minimum adjusted $p$-value $\leq 0.05$.

### Permissions and permits

All experiments in French Polynesia were conducted in accordance with relevant permissions (Protocole d'Accueil 2017–2021) from the Délégation à la Recherche and the Haut-commissariat de la République en Polynésie Française (DTRT). Coral samples utilized in microbiome analyses were exported from French Polynesia to the United States in accordance with CITES permit FR2298700091-E. The experiment on Palmyra Atoll was conducted by US Fish and Wildlife Service (USFWS) Special Use Permit (SUP) 12533-22023. All coral and sea cucumber species used in our experiments are invertebrates and did not require ethics approval.

### Reporting summary

Further information on research design is available in the Nature Portfolio Reporting Summary linked to this article.

## Data availability

The coral tissue mortality data generated in this study have been deposited in the BCO-DMO data system (https://www.bco-dmo.org/project/837802). Raw sequence data are available at NCBI's SRA Database under BioProject PRJNA1013970. All data needed to evaluate the conclusions in the paper are present in the paper and/or the Supplementary Information. Source data are provided in this paper.

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

## Acknowledgements

We thank the French Polynesian Government (Délégation à la Recherche) and the Haut-commissariat de la République en Polynésie Française (DTRT) for permits (Protocole d'Accueil 2017–2021) to work in French Polynesia and the US Fish and Wildlife Service and The Nature Conservancy for permits and support to work in Palmyra Atoll. N. Altman-Kurosaki, S. Bilodeau, and A. Caughman assisted in the field. Financial support came from the U.S. National Science Foundation (grant no. OCE 1947522; awarded to M.E.H. and F.J.S.), the National Geographic Society (grant no. NGS-57078R-19; awarded to M.E.H. and C.S.C.), the Teasley Endowment to the Georgia Institute of Technology (awarded to M.E.H.), and the Anna and Harry Teasley Gift Fund (awarded to M.E.H.). This work represents a contribution of the Moʻorea Coral Reef (MCR) LTER Site supported by the U.S. National Science Foundation (grant no. OCE 16–37396).

## Author contributions

C.S.C. and M.E.H. conceived the ideas and designed the study. C.S.C., Z.A.P. and M.E.H. collected the data. C.S.C. and Z.A.P. analyzed the data. C.S.C. and M.E.H. led the writing of the manuscript. C.S.C., Z.A.P., F.J.S. and M.E.H. contributed critically to the drafts and gave final approval for publication.

## Competing interests

The authors declare no competing interests.
