## [Peer Review File · Nature Communications]

Reviewers' Comments:

Reviewer #1:

Remarks to the Author:

This is an excellent contribution that very simply explains ecological shifts as drivers of coral diseases. The authors hypothesize that marine sediment may be a reservoir for coral diseases, showing that when a key species that plays a role in sediment turnover is removed, a particular type of coral disease (which affects fast growing Acroporid corals from the base) is in abundance. Further, authors show that sediment microbiomes differ in community composition as a function of sea cucumber presence or absence. This paper is incredibly significant for the understanding of several important fields of biology: 1) coral disease dynamics, which has been poorly understood from the beginning, 2) consequences of overfishing (sea cucumbers are over harvested in the regions studied, 3) ecology of key species and consequences of their removal and 4) these effects on marine microbial ecology/shifts.

The manuscript is exceptionally well-written, the abstract and introduction are appropriate, thorough and clear. Field methods are elegant (as is always true with this group of scientists) and microbial methods are very thorough. Images are clear. The discussion is compelling. This is a very important contribution which could influence policy change regarding fisheries and aid in an understanding of coral disease dynamics and other related fields. It is very appropriate for this journal. I honestly found nothing to suggest to improve upon.

Reviewer #2:

Remarks to the Author:

This paper is well written and presents some interesting data on under-appreciated interactions on coral reefs – the roles of sea cucumbers and farmer damselfish in moderating coral disease. The experiments are well executed and the results are generally compelling, and I expect this could be an influential paper that will prompt additional research on these interactions. My main concerns are that, as currently presented, the farmer damselfish aspect of the work feels like it is being short thrifted – I think this component of the work needs to be better integrated and highlighted. Additionally, the microbe community analysis somewhat undermines the narrative that sea cucumbers suppress sediment-derived pathogens, and yet that result is not highlighted in the abstract nor reflected in the framing of the paper. I expand on these thoughts for improvements below.

Major comments

- The title, abstract and introduction are written as if this is a story entirely about sea cucumbers modifying the prevalence of coral disease, with the exception of one sentence in the abstract about farmerfish that comes out of nowhere for the reader. Perhaps rather than focusing on sea cucumbers, the paper should focus on coral pathogens associated with sediments and how different components of the reef community (damselfish, sea cucumbers) influence coral exposure to these pathogens? Otherwise it feels like there is a disconnect between the story the authors are trying to tell and the data they are presenting. In particular, based on the results in Figure 2, turf algae on the coral base (resulting from damselfish farming) appears to be way more protective than sea cucumbers (tissue mortality reduced 94-100% by turf, vs 26-42% by sea cucumbers), and yet the paper's headlines are about sea cucumbers? The conclusion talks about cultivating and outplanting sea cucumbers to protect reefs, but there is no similar discussion of damselfish despite their stronger protective effect.

- Page 1, 3rd paragraph: The paper would benefit from more explanation and review of the role of sea cucumbers in controlling sediment-associated pathogens. I realize they are consuming a lot of sediment, but they are also pooping a lot of material back into the sediment. Do some of these pathogens not persist in sea cucumber feces? Are there certain pathogens that are facilitated by sea cucumber poop? Would we expect a linear relationship between sea cucumber density and pathogen prevalence, or might it be a humped shaped relationship where intermediate levels of sea cucumbers lead to the lowest pathogen levels?

- Page 2, first sentence: "mining of lootable resources," even if it is a quote, is value laden language that is unnecessary. Also, while I agree that many sea cucumber fisheries are

unsustainable, causes of unsustainable harvest include extreme poverty, lack of management measures, and lack of enforcement. It is certainly possible to have a sustainable sea cucumber fishery. This sentence reads as if all sea cucumber fisheries are "bad."

- The fact that the microbes that increased in sediments without sea cucumbers were NOT the microbes that were common in diseased coral samples really complicates the story presented in this paper. The data clearly show that sea cucumbers seem to play a role in mediating coral disease, but the mechanism is unclear from the data presented. Because of this, and the lack of a more detailed exploration of the role of different sea cucumber densities on coral disease, this study feels somewhat preliminary.

- When discussing the results, I think the authors should explore how their results might vary if sea cucumber densities were at historical, unfished levels. They cite densities of up to 50 individuals per m² prior to large-scale harvesting, which is much higher than their experimental densities.

Other comments

- Last sentence on page 1: This estimate isn't quite current (looks like data are from 2004-2008) and it is based on data from a subset of countries. I think this sentence needs some qualifications to be more accurate.

- Page 2, line 32: I wouldn't use the word "consistently" for just two comparisons. I would change to "in both experiments." Sure, the results are consistent with one another, so the word isn't technically wrong, but it implies a broader pattern than what is presented here.

Reviewer #3:

Remarks to the Author:

The study by Clements and colleagues explores the relationship between holothurian presence and coral survival, considering the hypothesis that detritivorous holothurians may alter the composition of sediments and thus the effect of sediments on corals. Through a collection of experiments the authors conclude that coral fragments that are in direct contact with sediments in the absence of detritivory experience higher rates of tissue loss than do fragments either not in contact with sediments or in contact with sediments that are exposed to holothurian activities. The authors present results of additional treatments that introduce distance between coral tissue and sediments, including selecting coral fragments with turf algae adjacent to coral tissue. The results reinforce the conclusion that corals in contact with unforaged sediments suffer higher rates of tissue loss than other treatments. The authors complement these findings with microbiome assessments that suggest that the microbial community of sediments are different based upon levels of holothurian alteration and that microbial communities differ with proximity to sediments. The authors find no evidence of particular pathogens affecting tissue loss though explore a couple of causative pathways of microbial interactions with corals. The authors interpret these results through a lens of potential impacts of overharvest of holothurians to coral and coral reef fates.

The study introduces some new data with some specific experimental conditions. The results are consistent with the expectation that coral microbiomes can be significantly altered through direct contact with microbially-rich neighbors. Such expectations are among the hypotheses for why many reef-building corals have colonial structures that are elevated and minimize contact with the benthic substrate underneath the colony. The experiments presented here present data suggesting that bioturbation by holothurians can alter characteristics of the sediment microbial community, resulting in shifted impact of contact with coral colonies. This result is new based upon my reading of the literature of coral reef ecology.

A number of questions emerge, however, in trying to understand the details of the study. I will treat each major question individually.

Study design. As presented, the design appears to be haphazard. The study is conducted on two islands, which suggests an interest in testing the potential generality of results. However, the studies did not follow the same methods on each island. Perhaps the study was stepwise and incremental — general effects of holothurians (Figure 1c&d; Moorea), mechanistic view of holothurians through a cross with direct contact (Figure 1e; Palmyra), then alternative mechanism

of contact (Figure 2; Moorea). There are details of the experimental designs that are difficult to follow (e.g., the cages appear to be different across the three, there is no description of how corals were collected for Fig 1e). Further, the results from the two experiments on Moorea are not consistent. Figure 1c shows strong difference in tissue death as a function of holothurian presence (fragments in contact with sediment) while in Fig 2 the two treatments without turf, both in contact with the sediments, show no post hoc support for differences across holothurian treatment levels. Perhaps this is due to a significant difference in the experimental design, timeline, or measurement, but it is not apparent what this difference is. Some clarity is needed in the design with a reconciliation of inconsistent results across experiments.

Statistical analysis. There needs to be more clarity in the data presentations. The major results presented in the Abstract and elsewhere are imprecise. There is no reporting of error or confidence in the summary statistics, such as the reported 1500% difference in mortality. Further, this 15-fold difference in survivorship is explicitly linked to the duration of the study rather than being an estimate of instantaneous rate. As such, the reporting is lacking in clarity and precision, which limits the value of such summary presentations. In the analyses, it is unclear how mean and error are calculated. Are the statistics calculated with the independent replicate being the enclosure (or patch) or estimated on a per-colony basis (the latter invoking concerns of pseudoreplication)? Lines 436-437 seem to suggest that patch is treated as a random factor rather than as the replicate; I would consider fragments to be nested within patch, again with the understanding that patches are the independent replicate unit. Additionally, I would expect more transparency to the descriptive statistics. On Lines 409 and 444, for example, the estimates of holothurian density should be reported more formally, namely with mean and some metric of variance (beyond range) presented. As a study of the impacts of holothurians, it is critical to know whether such organisms are patchy across the reefscape or are fairly homogeneously distributed. Such information is even more important given that the experiments were conducted in two locations.

Sampling rationale. What is the rationale for sacrificing all corals once only one coral suffers >50% mortality (see Line 482)? Following this approach, there is disproportionate weighting offered to extreme individuals. Such individual fragments may die for any host of reasons, including ones that have nothing to do with the treatment. By sacrificing all fragments at different points in time and with different levels of mortality within a patch, there is a host of variability introduced to the data. If one patch is collected at Day 7 while another is collected at Day 30, there are many reasons why the microbial associates might differ in ways that are independent of the experimental treatments. Additionally, there are no data reported regarding the variation in coral colony condition at the time of collection. The data were to be interpreted much differently were all colonies to be suffering similar levels of mortality (e.g., all near to 50% tissue loss) than were there to be only one colony with >50% mortality and the rest being 100% intact. Please provide a justification for the approach and some transparency of the variability within patch that will help to contextualize the results.

Interpretation. While I appreciate the goal to link experimental treatments to broader environmental considerations, I find that the contextualization of this study to be a bit extreme. The results presented here provide a fairly direct reporting of small-scale effects of bioturbation by holothurians. The effects of holothurians are context-specific, likely affecting a subset of corals that live in sediment-rich habitats and have high rates of contact with sediments. As alluded to earlier, growth morphologies of corals reveal geometries that limit contact with neighboring substrates, including benthic sediments. This study explored the conditions of small coral fragments placed directly in sediments, a condition affecting a subset of corals during fragmentation or early life history. Certainly these phases of corals can have demographic impact, especially during post-disturbance times of recovery. The manuscript, however, draws some broad links to patterns of disease and holothurians, which seems hyperbolic given the natural history of corals. This discussion may be viewed as further extreme given that no specific disease agents were identified, so even invoking the term "disease" requires some important definitions of the usage of the term.

The study has some intriguing results that contribute to our understanding of the role played by holothurians in bioturbation and potential impacts on other reef taxa. With clarifications of the study design and implementation, along with some sincere contextualization of the results, the work will contribute to our expanding understanding of coral reef ecology.

Reviewer #4:

Remarks to the Author:

General comments

This study focuses on the role of sediment-dwelling microbes facilitating the infection and death of reef dwelling acroporid corals. Sediments harboring the pathogens were rendered less infectious by deposit feeding sea cucumbers. Where sea cucumbers were absent but territorial damselfish were present (called here "farmerfish") infection rates declined. This was because fish predation removed coral tissue near the sediment-coral interface sufficiently to isolate the rest of the colony from the pathogenic sediments.

Overall, I found the research and analyses supported the key points. Illustrations were clear and striking. Statistical analyses were appropriate and support all the key conclusions.

The current rarity and economic value of sea cucumbers is well known. Prior to this research most researchers, managers and policy makers thought sea cucumbers were simply ecosystem passengers rather than ecosystem drivers. The emerging story suggests this group may play a critical role in the population dynamics of reef coral.

This study employed elegant manipulative experiments that altered sea cucumber population densities and distributions. I fully appreciated the use of historical data to paint the picture of abundant sea cucumbers prior to escalating harvesting efforts.

I've been trying to find weaknesses I can identify but have come up short. However, Figs 3 A, C and D, have three colors to identify the coral outplant experiment. However, some colorblind readers will not be able to distinguish colors among colored points.

This is an important study. Diseases are proliferating among reef corals throughout the tropics but most managers and policy makers feel these are unmanageable. This study suggests a path forward that could curb agents of coral disease transmission.

Finally, this study could have been hopelessly complicated. It includes sediment microbiomes, echinoderm populations, coral, fisheries and disease impacts in geographically broad experimental manipulative studies. It would be very easy to have readers lost in all those weeds. However, I found the paper beautifully written for its clarity and voice. Finally, I appreciate the authors integrating this marine study into the larger picture of terrestrial detritivores. This is often lacking in "marine ecology" studies but it is an important conceptual link that will broaden the pool of readers of this study.

Specific comments:

Line 22:

Farmerfish are territorial damselfish. However, their impacts seem to me to be at distal rather than basal areas of branched acroporids. Is there evidence of the former?

Figure 3. Colorblind readers cannot detect differences in the three-color pattern (e.g., Fig. 3D).

Below we have pasted the reviewer's comments along with our responses, which are in blue.
REVIEWER COMMENTS

Reviewer #1 (Remarks to the Author):

This is an excellent contribution that very simply explains ecological shifts as drivers of coral diseases. The authors hypothesize that marine sediment may be a reservoir for coral diseases, showing that when a key species that plays a role in sediment turnover is removed, a particular type of coral disease (which affects fast growing Acroporid corals from the base) is in abundance. Further, authors show that sediment microbiomes differ in community composition as a function of sea cucumber presence or absence. This paper is incredibly significant for the understanding of several important fields of biology: 1) coral disease dynamics, which has been poorly understood from the beginning, 2) consequences of overfishing (sea cucumbers are over harvested in the regions studied, 3) ecology of key species and consequences of their removal and 4) these effects on marine microbial ecology/shifts.

The manuscript is exceptionally well-written, the abstract and introduction are appropriate, thorough and clear. Field methods are elegant (as is always true with this group of scientists) and microbial methods are very thorough. Images are clear. The discussion is compelling. This is a very important contribution which could influence policy change regarding fisheries and aid in an understanding of coral disease dynamics and other related fields. It is very appropriate for this journal. I honestly found nothing to suggest to improve upon.

Thanks for the kind words. We did not edit in response to this reviewer's comments.

Reviewer #2 (Remarks to the Author):

This paper is well written and presents some interesting data on under-appreciated interactions on coral reefs – the roles of sea cucumbers and farmer damselfish in moderating coral disease. The experiments are well executed and the results are generally compelling, and I expect this could be an influential paper that will prompt additional research on these interactions. My main concerns are that, as currently presented, the farmer damselfish aspect of the work feels like it is being short thrifted – I think this component of the work needs to be better integrated and highlighted. Additionally, the microbe community analysis somewhat undermines the narrative that sea cucumbers suppress sediment-derived pathogens, and yet that result is not highlighted in the abstract nor reflected in the framing of the paper. I expand on these thoughts for improvements below.

We agree that emphasizing the farmerfish effect and adding more discussion on the microbiome component is useful. We have done this in the main text (and will give the specifics of this below where the reviewer has more focused comments). However, in the abstract it is not possible to cover these additional topics within the allowable word limit so we have focused on the major findings only and will give more context in the main text. In the previous abstract (for Nature) we had 184 words, cutting it to 150 further limited our ability to cover these multiple outcomes in the Abstract.

Major comments

- The title, abstract and introduction are written as if this is a story entirely about sea cucumbers modifying the prevalence of coral disease, with the exception of one sentence in the abstract about farmerfish that comes out of nowhere for the reader.

We have added text in the Introduction to better introduce the effects of the farmerfish. On lines 61-77 in the introduction we now note:

*“An additional consideration is that coral diseases associated with sediment exposure often seem to occur following direct contact with sediment or turbulence that moves sediment onto corals⁷⁻¹¹. If direct contact risks infection, then species, like some *Acropora*, that form thickets directly on sand should be at especially high risk. However, once thickets are established, their bases are commonly infested with damselfishes that kill coral bases, cultivate algal gardens on these bases²⁷, and thus separate live coral tissues from sediment contact via a barrier of filamentous turf algae. These farmerfishes appear to be parasites that kill coral bases to grow the algal gardens from which they feed, but it is possible that in the larger community context, they function as mutualists, protecting the coral from the dangers of disease in the sediment below the coral.*

*Here we show that two *Acropora* species associated with sandy environments on both a high island with a human population and a low island (atoll) with no human population experience lower frequencies of disease and greater survivorship when sea cucumbers are present then when they are removed. Additionally, even without sea cucumbers cleaning sediments, once corals grow to sizes that attract farmerfishes that cultivate filamentous algal gardens on coral bases, the algae separating live coral tissue from contact with sediment strongly protects corals from infection.”*

Perhaps rather than focusing on sea cucumbers, the paper should focus on coral pathogens associated with sediments and how different components of the reef community (damselfish, sea cucumbers) influence coral exposure to these pathogens? Otherwise it feels like there is a disconnect between the story the authors are trying to tell and the data they are presenting.

The farmerfish won't aid coral in recruiting via larvae, in early juvenile survival, or when vegetative fragments are dispersed across sediments (a common mode of spread for these types of *Acropora* because the farmerfish don't colonize the coral patch until it is large enough to offer both shelter and adequate basal space for culturing their algal gardens. Thus, the suppression of disease by sea cucumbers is critical for reef resilience following disturbances such as bleaching events, crown-of-thorns outbreaks, cyclone damage, etc. when recovery must occur via recruitment or surviving fragments. We now note this in the lower paragraph of the above statement and also on lines 245-252 in the Results and Discussion where we say:

*“For corals like *A. pulchra* (or *A. cervicornis* in the Caribbean) that can form thickets on sandy sediment and commonly spread via fragmentation, our outplants in contact with sediments would mirror the rigors of fragmentation and vegetative spread. Our data suggest that sea cucumbers can provide critical benefits by preventing death due to disease during this colonization stage. Once colonizers persist and grow to a size*

sufficient to offer shelter and substrate to farmerfishes, the advantages of sea cucumbers appear less critical. The beneficial effect of farmerfish gardens in separating live coral tissues from direct contact with sediment allows continued growth and spread of the patch even without sea cucumbers nearby.”

In particular, based on the results in Figure 2, turf algae on the coral base (resulting from damselfish farming) appears to be way more protective than sea cucumbers (tissue mortality reduced 94-100% by turf, vs 26-42% by sea cucumbers), and yet the paper’s headlines are about sea cucumbers? The conclusion talks about cultivating and outplanting sea cucumbers to protect reefs, but there is no similar discussion of damselfish despite their stronger protective effect.

For corals to get established, they have to first colonize and then grow to sufficient size to make an adequately branched and larger “patch” that can harbor damselfishes and support their algal gardens. Thus, the damsels help with persistence once the patch is established, but cannot be “outplanted” (as can sea cucumbers) and persist until the coral is already there to provide the required habitat. Additionally, damselfishes are not harvested and are abundant on most reefs (i.e., not in need of additional human efforts) – in contrast, sea cucumbers are harvested and due to that are now rare. The last paragraph in the response above explains the role that sea cucumbers play in establishment and that farmerfishes play in persistence once established.

- Page 1, 3rd paragraph: The paper would benefit from more explanation and review of the role of sea cucumbers in controlling sediment-associated pathogens. I realize they are consuming a lot of sediment, but they are also pooping a lot of material back into the sediment. Do some of these pathogens not persist in sea cucumber feces? Are there certain pathogens that are facilitated by sea cucumber poop? Would we expect a linear relationship between sea cucumber density and pathogen prevalence, or might it be a humped shaped relationship where intermediate levels of sea cucumbers lead to the lowest pathogen levels?

These are excellent questions but not ones that are presently answerable. First, much of the sea cucumber literature has focused on the fishery aspect, trying to document recovery (or lack thereof) following harvest, or on determining general levels of “organics” in sediment going into versus coming out of sea cucumbers – careful studies of changes in the microbial community as sediment is passed through sea cucumber guts are not available. Second, even if these were available (and we make some evidence available here in terms of microbial community composition of sediments protected from versus grazed by sea cucumbers), very few coral pathogens have been rigorously identified so neither we, nor other researchers, really know what to look for in terms of known pathogens. Third, in several coral diseases (black band being one good example), the “deathfront” that moves across the coral is a polyspecies mix of bacteria and it is unclear which are the pathogens and which are opportunists taking advantage of the compromised coral tissue that has died due to other actors (i.e., some components may be assassins, but some are just vultures, and it is not clear which are which). We cover this in more detail in response to a later comment and have edited the main text to address some of these issues (see lines 205-219).

- Page 2, first sentence: “mining of lootable resources,” even if it is a quote, is value laden language that is unnecessary. Also, while I agree that many sea cucumber fisheries are

unsustainable, causes of unsustainable harvest include extreme poverty, lack of management measures, and lack of enforcement. It is certainly possible to have a sustainable sea cucumber fishery. This sentence reads as if all sea cucumber fisheries are “bad.”

The reviewer is correct that some sea cucumber fisheries may be sustainable, especially where small island groups harvest for food and don't overharvest. The vast majority of harvest is for the Asian market (not for sustainable local use), with vast collections being made that tend to clearly overfish first areas, then regions, then even larger areas. In recent decades harvest has progressed from shallow to deep waters, from a focus on high value species to a focus on species of lesser value, and has been forced to move into new regions, countries, and ocean basins due to depletion of the resource. We have retained the quote because much of the fishing has been conducted like mining – collecting all that fishers can find and then moving on to new areas. In the text on lines 44-56 we note:

“However, sea cucumbers have been heavily harvested since the 1800s and are now functionally extirpated from many reefs^{15,16}. World-wide harvests increased dramatically in the 1960s¹⁷ and are continuing to increase; during the 2011-2020 period, annual wild harvests increased by ~30% and reached 57,700 tonnes of dried sea cucumbers¹⁸. This tonnage can be estimated to represent a harvest of more than 1,000,000,000 individuals/yr. Because recovery following harvest takes decades, or fails entirely, sea cucumber harvest is commonly a “mining of lootable resources” rather than a sustainable fishery^{15,19}, with harvests moving from shallower to deeper waters, from high value to lower value species, and into new geographic regions as previously fished regions are depleted^{17,18}. Reliable estimates of historic sea cucumber densities are unavailable due to past large-scale harvesting, but densities of up to 50 individuals m⁻² were reported in some remote locations in the 1960s²⁰⁻²². Their value as a food, ease of harvest, and inefficient reproduction at low densities led to the widespread collapse of populations¹⁶, even with management intervention²³.

- The fact that the microbes that increased in sediments without sea cucumbers were NOT the microbes that were common in diseased coral samples really complicates the story presented in this paper. The data clearly show that sea cucumbers seem to play a role in mediating coral disease, but the mechanism is unclear from the data presented. Because of this, and the lack of a more detailed exploration of the role of different sea cucumber densities on coral disease, this study feels somewhat preliminary.

We agree that this is less satisfying than we had hoped, BUT we were not especially surprised given other efforts to identify the agents causing coral disease. To better inform a reader, we now note on lines 205-219:

*“The lack of a clear spike of a suspected pathogen may not be surprising if the causative microorganism(s) is a natural component of the environmental or host microbiome and only becomes pathogenic if circumstances allow. This is the nature of opportunistic pathogens, for example various *Enterococcus* spp. that are commensal in the human GI tract but disease-causing if disturbance (e.g., wounding, ulcers, surgeries) allows spread to other body sites⁴³. Such scenarios could also involve communities of microorganisms (a polymicrobial consortium), analogous to what has been suggested in black band disease,*

with host damage driven by diverse metabolic outcomes associated with microbial overgrowth⁴⁴. Disease-causing microorganisms may also be hard to identify if they occur at low frequency but exert a pathogenic effect indirectly by "remodeling" a commensal microbiome into a harmful one, for example through changes in cell number, relative abundance, or metabolic activity⁴⁵. These so-called "keystone pathogens" are implicated in human consortia diseases (e.g., periodontitis) and have been hypothesized as agents of coral disease⁴⁰. Additional causes could include ciliates^{41,42}, fungi⁴⁶, or chemical changes at the sediment-coral juncture as microbes or organics accumulate in the sediment⁴⁷ without being removed or bioturbated by sea cucumbers."

- When discussing the results, I think the authors should explore how their results might vary if sea cucumber densities were at historical, unfished levels. They cite densities of up to 50 individuals per m² prior to large-scale harvesting, which is much higher than their experimental densities.

The reviewer's request is reasonable, but we can't say much in a data-based way on this due to so little being known about 1) the processing sea cucumbers do and the microbes consumed vs passed through and 2) "normal" pre-harvest densities. Despite historic accounts of 50m⁻² in some locations, we would be uncomfortable arguing that such densities were means, medians, usual, unusual, etc. The frustrating bottom-line is that historic densities are just not clear. We are reluctant to use those "fuzzy numbers" to elaborate a larger discussion of possibilities without a better foundation for historic densities and for which microbes are digested versus passed through. We are working on some aspects of the latter issue and hope this publication stimulates others to do the same.

Other comments

- Last sentence on page 1: This estimate isn't quite current (looks like data are from 2004-2008) and it is based on data from a subset of countries. I think this sentence needs some qualifications to be more accurate.

Thanks for pointing this out. We now cite the 30% increase in harvest between 2011 and 2020 and the total mass of 57,700 tonnes of annual harvest given by Percell et al. (2023).

- Page 2, line 32: I wouldn't use the word "consistently" for just two comparisons. I would change to "in both experiments." Sure, the results are consistent with one another, so the word isn't technically wrong, but it implies a broader pattern than what is presented here.

Fair enough. This has been altered as suggested – on lines 106-110, we say, "*Thus, despite evaluating different sea cucumbers, different corals, a high island populated by humans and a low island with minimal human impact, and different sites separated by ~3,000 km, sediment-feeding sea cucumbers suppressed coral tissue death associated with sediment contact in both locations.*".

Reviewer #3 (Remarks to the Author):

The study by Clements and colleagues explores the relationship between holothurian presence

and coral survival, considering the hypothesis that detritivorous holothurians may alter the composition of sediments and thus the effect of sediments on corals. Through a collection of experiments the authors conclude that coral fragments that are in direct contact with sediments in the absence of detritivory experience higher rates of tissue loss than do fragments either not in contact with sediments or in contact with sediments that are exposed to holothurian activities. The authors present results of additional treatments that introduce distance between coral tissue and sediments, including selecting coral fragments with turf algae adjacent to coral tissue. The results reinforce the conclusion that corals in contact with unforaged sediments suffer higher rates of tissue loss than other treatments. The authors complement these findings with microbiome assessments that suggest that the microbial community of sediments are different based upon levels of holothurian alteration and that microbial communities differ with proximity to sediments. The authors find no evidence of particular pathogens affecting tissue loss though explore a couple of causative pathways of microbial interactions with corals. The authors interpret these results through a lens of potential impacts of overharvest of holothurians to coral and coral reef fates.

The study introduces some new data with some specific experimental conditions. The results are consistent with the expectation that coral microbiomes can be significantly altered through direct contact with microbially-rich neighbors. Such expectations are among the hypotheses for why many reef-building corals have colonial structures that are elevated and minimize contact with the benthic substrate underneath the colony. The experiments presented here present data suggesting that bioturbation by holothurians can alter characteristics of the sediment microbial community, resulting in shifted impact of contact with coral colonies. This result is new based upon my reading of the literature of coral reef ecology.

A number of questions emerge, however, in trying to understand the details of the study. I will treat each major question individually.

Study design. As presented, the design appears to be haphazard. The study is conducted on two islands, which suggests an interest in testing the potential generality of results. However, the studies did not follow the same methods on each island. Perhaps the study was stepwise and incremental — general effects of holothurians (Figure 1c&d; Moorea), mechanistic view of holothurians through a cross with direct contact (Figure 1e; Palmyra), then alternative mechanism of contact (Figure 2; Moorea). There are details of the experimental designs that are difficult to follow (e.g., the cages appear to be different across the three, there is no description of how corals were collected for Fig 1e).

Regarding coral collections in Mo'orea, on lines 141-144 we say:

“Coral portions for the experiment were collected from numerous A. pulchra patches in the area near our experimental cages, and this experiment was monitored daily for 36 days, visually estimating % tissue death of each coral each day.”

For coral collections on Palmyra Atoll, we say on lines 315-316:

“Coral portions for the experiment were collected from 15 A. nasuta colonies in the East Lagoon of Palmyra Atoll where our experimental cages were located.”

Regarding the differing experimental approaches on each island, we now explain this on lines 304-307 where we note:

“We did not perform large-scale removals (as we did in Mo’orea) due to Palmyra Atoll being a reserve managed by the US Fish and Wildlife Service and their permitting process limits such large-scale manipulations when more modest manipulations (in this case smaller cages) can address the question posed.”

Other than that difference, the caging efforts used similar sized cages on both islands and there were no cages in the large-scale removal plots in the initial experiment on Mo’orea.

Further, the results from the two experiments on Moorea are not consistent. Figure 1c shows strong difference in tissue death as a function of holothurian presence (fragments in contact with sediment) while in Fig 2 the two treatments without turf, both in contact with the sediments, show no post hoc support for differences across holothurian treatment levels.

We are confused by this statement. There is a significant effect of sea cucumber presence in the data presented in Figure 2, with those exposed to sea cucumbers experiencing less damage. The effect size and p-value ($p = 0.001$) was greater for corals with algal turf on their base, but the effect of sea cucumbers was also significant ($p = 0.020$) and there was no significant turf x sea cucumber interaction. If the reviewer is noting that the post-hoc test for no-turf w/ sea cucumbers does not differ significantly from the effects on contact-corals w/o sea cucumbers, it is my understanding (and experience) that post-hoc tests have less power than the full Permutation-based LME model and that it is not rare for the full model to detect differences that cannot be documented in all cases in a post-hoc evaluation. There may also be some effects of duration here. The large-scale removal experiment ran for 45 days while the caging study ran for only 36 days. We often see greater differences among treatments emerge as time progresses, but travel schedules, lab costs, and availability of living and lab space at the field station in Mo’orea sometimes dictate experimental durations.

Perhaps this is due to a significant difference in the experimental design, timeline, or measurement, but it is not apparent what this difference is. Some clarity is needed in the design with a reconciliation of inconsistent results across experiments.

As above, we do get a significant effect of sea cucumber presence in all experiments, so we are unsure of what is needed here.

Statistical analysis. There needs to be more clarity in the data presentations. The major results presented in the Abstract and elsewhere are imprecise. There is no reporting of error or confidence in the summary statistics, such as the reported 1500% difference in mortality. Further, this 15-fold difference in survivorship is explicitly linked to the duration of the study rather than being an estimate of instantaneous rate. As such, the reporting is lacking in clarity and precision, which limits the value of such summary presentations.

For % tissue death or other metrics that give a range of values, we can, and do, report means and variances. For survival analyses where the values are categorical (either alive or dead), we are unaware of any metric to report variance. We have, however, added a bit more information. The relevant metric for survival analysis is the hazard ratio, which is now included on lines 89-91. The hazard ratio is often used to express an increase or reduction in the risk of an event occurring (Although there is some nuance around the semantics of this concept. See the link to Sashegyi and Ferry 2017 below), for example where they say,

“It is a common practice when reporting results of cancer clinical trials to express survival benefit based on the hazard ratio (HR) from a survival analysis as a “reduction in the risk of death,” by an amount equal to $100 \times (1 - \text{HR}) \%$. Stating, for instance, that “drug X reduces the risk of dying by 40%,” based on an observed survival HR of 0.60, is a typical way of communicating survival benefit.”

Furthermore, this hazard ratio is an estimate of an instantaneous rate:

“The “hazard” is an instantaneous, as opposed to a cumulative, risk. In lay terms, the hazard of an event at some time point t may be thought of as the chance of that event occurring at time t , given event-free survival up to t ...”

Sashegyi and Ferry (2017) On the Interpretation of the Hazard Ratio and Communication of Survival Benefit. *Oncologist* 22: 484–486

We assume that most readers will better understand the statement of a 1500% increase in mortality and that fewer are conversant with the meaning of the hazard ratio but to be as clear as possible, we now say on lines 89-91:

“Whole colony mortality was 1500% greater (30% vs 2%; Survival Analysis, Hazard ratio 0.056 ± 0.060 , $p = 0.007$, Fig. 1d). Thus, sea cucumber presence within patches reduces the risk of whole colony mortality by ~94%..”

In the analyses, it is unclear how mean and error are calculated. Are the statistics calculated with the independent replicate being the enclosure (or patch) or estimated on a per-colony basis (the latter invoking concerns of pseudoreplication)? Lines 436-437 seem to suggest that patch is treated as a random factor rather than as the replicate; I would consider fragments to be nested within patch, again with the understanding that patches are the independent replicate unit.

There are 5 measurements per patch that each derived from 5 separate corals that were in that respective patch. Technically, these 5 measurement units are pseudo replicates—the patches themselves are the experimental units ($n = 10$ per treatment), but we use a mixed effects model to account for pseudoreplication by specifying patch as a random factor in the model, as described in our methods:

On lines 291-298, we note:

“We used permutation-based, linear mixed-effects (LME) model in the R⁵⁷ package predictmeans⁵⁸ to compare differences in percent tissue mortality between corals in plots where sea cucumbers were removed vs. not removed or in contact or not with sediment. Patch type (removed vs. not removed) was treated as a fixed factor, with replicate patches treated as a random effect nested within patch type. To compare overall coral survivorship at 45 days, survival curves and analyses were generated using random-effects Weibull regression in Stata (version 17), with patch type as a fixed effect and replicate patches treated as a random effect nested within patch type.”

For more information on experimental units vs. measurement units, and dealing with pseudoreplication via mixed-effect models, please see: VSNi – Dealing with Pseudoreplication in Linear Mixed-Effect Models

Additionally, I would expect more transparency to the descriptive statistics. On Lines 409 and 444, for example, the estimates of holothurian density should be reported more formally, namely with mean and some metric of variance (beyond range) presented. As a study of the impacts of holothurians, it is critical to know whether such organisms are patchy across the reefscape or are fairly homogeneously distributed. Such information is even more important given that the experiments were conducted in two locations.

We have provided the mean \pm SE density of holothurians for our study sites in Mo’orea (lines 238-239) and Palmyra Atoll (line 308).

Sampling rationale. What is the rationale for sacrificing all corals once only one coral suffers >50% mortality (see Line 482)? Following this approach, there is disproportionate weighting offered to extreme individuals. Such individual fragments may die for any host of reasons, including ones that have nothing to do with the treatment. By sacrificing all fragments at different points in time and with different levels of mortality within a patch, there is a host of variability introduced to the data. If one patch is collected at Day 7 while another is collected at Day 30, there are many reasons why the microbial associates might differ in ways that are independent of the experimental treatments. Additionally, there are no data reported regarding the variation in coral colony condition at the time of collection.

Actually, some data are reported and the strength of effect on certain treatments and not others is strong. On lines 148-152 we note:

*“Other corals in that cage were sampled 1 cm from the tip (apical samples) and at the lowest portion of live coral tissue so that microbiomes of corals contacting sediment **(100% were diseased)** versus separate from sediment by turf algae **(only 3% were diseased)** from the same cage and time could be compared via Illumina sequencing.”*
(Bold and color added here to point out the magnitude of this difference and the data presented.)

The data were be interpreted much differently were all colonies to be suffering similar levels of mortality (e.g., all near to 50% tissue loss) than were there to be only one colony with >50%

mortality and the rest being 100% intact. Please provide a justification for the approach and some transparency of the variability within patch that will help to contextualize the results.

There is often considerable spatial variance (or variance in clone susceptibility?) that occurs in these experiments – collecting when one fragment in a replicate experiences 50% death allows us to assess RELATIVE effects at a location w/o confounding location with treatment effects (since all treatments were spatially blocked within each independent replicate). Had we waited a uniform time and collected all replicates – say at day 20 – different blocks of treatments at several sites may have all been dead while blocks of treatments at other sites may have all been fine. In these and other experiments, we see considerable spatial variance in infection rate – whether this is due to spatially patchy grazing by sea cucumbers, patchy distribution of the unidentified pathogen, spatial differences in physical factors (a fish popped there or did not, etc.), we don't know, but this design allows for minimizing that spatial variance. On lines 152-159 we now say:

“Because there is spatial variance in timing of disease onset, this sampling scheme allowed i) all treatments at a small-scale site (the cage replicate, which served as a block for the three outplant types) to be sampled synchronously and minimize effects of spatial as opposed to treatment variance and ii) assessing the coral microbiome for possible identification of the disease-causing organism at the site of the moving white-band of disease. Had we sampled all replicates at a uniform time, diseased treatments in some replicates would have progressed to complete mortality while no outplants in others would have been infected -making sampling at the disease front impossible for many replicates.”

Interpretation. While I appreciate the goal to link experimental treatments to broader environmental considerations, I find that the contextualization of this study to be a bit extreme. The results presented here provide a fairly direct reporting of small-scale effects of bioturbation by holothurians. The effects of holothurians are context-specific, likely affecting a subset of corals that live in sediment-rich habitats and have high rates of contact with sediments. As alluded to earlier, growth morphologies of corals reveal geometries that limit contact with neighboring substrates, including benthic sediments. This study explored the conditions of small coral fragments placed directly in sediments, a condition affecting a subset of corals during fragmentation or early life history. Certainly these phases of corals can have demographic impact, especially during post-disturbance times of recovery. The manuscript, however, draws some broad links to patterns of disease and holothurians, which seems hyperbolic given the natural history of corals. This discussion may be viewed as further extreme given that no specific disease agents were identified, so even invoking the term “disease” requires some important definitions of the usage of the term.

- 1) In some respects, we agree with the reviewer here. These effects will fall more heavily on coral associated with sandy environments and we have noted in the text that we are working with two such species. We have also now noted the especially critical role these interactions may play in recruitment and recovery post disturbance. On lines 245-252 we now say:

“For corals like A. pulchra (or A. cervicornis in the Caribbean) that can form thickets on sandy sediment and commonly spread via fragmentation, our

outplants in contact with sediments would mirror the rigors of fragmentation and vegetative spread. Our data suggest that sea cucumbers can provide critical benefits by preventing death due to disease during this colonization stage. Once colonizers persist and grow to a size sufficient to offer shelter and substrate to farmerfishes, the advantages of sea cucumbers appear less critical. The beneficial effect of farmerfish gardens in separating live coral tissues from direct contact with sediment allows continued growth and spread of the patch even without sea cucumbers nearby.”

- 2) However, it is not clear that we are overstating our findings or that similar effects do not occur in hard substrate environments on reefs subject to some sediment mobility during storms, etc. In the Florida Keys, there have been several tens of thousands of *Acropora cervicornis* that were grown in near-shore nurseries (suspended above sediments) and then outplanted onto hard substrates on the barrier reef. Although there are a few locations experiencing some success, in the majority of cases, these corals persist and grow for a while, then develop a whiteband of infection at the base that works its way up the colony and kills most outplants. We do not know that the disease agents there are the same as we see here, but the symptoms are the same.
- 3) Finally, the fact that we see similar effects of sea cucumber feeding on coral health using a high populated island and a low unpopulated island, two different corals, two different sea cucumbers, and study sites separated by ~3,000km suggests a robustness to the pattern.
- 4) Regarding “no specific disease agents were identified” – We agree that this is frustrating and we are now involved in additional studies trying to identify the pathogen(s), but this situation is very common in the coral disease literature. For most coral diseases, indeed most marine diseases (see Vega-Thurber and Hay 2023, Science Advances), the causative agents remain unknown. We now address this on lines 205-218 by noting:

*“The lack of a clear spike of a suspected pathogen may not be surprising if the causative microorganism(s) is a natural component of the environmental or host microbiome and only becomes pathogenic if circumstances allow. This is the nature of opportunistic pathogens, for example various *Enterococcus* spp. that are commensal in the human GI tract but disease-causing if disturbance (e.g., wounding, ulcers, surgeries) allows spread to other body sites⁴³. Such scenarios could also involve communities of microorganisms (a polymicrobial consortium), analogous to what has been suggested in black band disease, with host damage driven by diverse metabolic outcomes associated with microbial overgrowth⁴⁴. Disease-causing microorganisms may also be hard to identify if they occur at low frequency but exert a pathogenic effect indirectly by "remodeling" a commensal microbiome into a harmful one, for example through changes in cell number, relative abundance, or metabolic activity⁴⁵. These so-called "keystone pathogens" are implicated in human consortia diseases (e.g., periodontitis) and have been hypothesized as agents of coral disease⁴⁰. Additional causes could include ciliates^{41,42}, fungi⁴⁶, or chemical changes at the sediment-coral juncture*

as microbes or organics accumulate in the sediment⁴⁷ without being removed or bioturbated by sea cucumbers.”

The study has some intriguing results that contribute to our understanding of the role played by holothurians in bioturbation and potential impacts on other reef taxa. With clarifications of the study design and implementation, along with some sincere contextualization of the results, the work will contribute to our expanding understanding of coral reef ecology.

We agree and thank the reviewer for the above comments. Responding to these issues has improved the manuscript and we are grateful for the input.

Reviewer #4 (Remarks to the Author):

General comments

This study focuses on the role of sediment-dwelling microbes facilitating the infection and death of reef dwelling acroporid corals. Sediments harboring the pathogens were rendered less infectious by deposit feeding sea cucumbers. Where sea cucumbers were absent but territorial damselfish were present (called here "farmerfish") infection rates declined. This was because fish predation removed coral tissue near the sediment-coral interface sufficiently to isolate the rest of the colony from the pathogenic sediments.

Overall, I found the research and analyses supported the key points. Illustrations were clear and striking. Statistical analyses were appropriate and support all the key conclusions.

The current rarity and economic value of sea cucumbers is well known. Prior to this research most researchers, managers and policy makers thought sea cucumbers were simply ecosystem passengers rather than ecosystem drivers. The emerging story suggests this group may play a critical role in the population dynamics of reef coral.

This study employed elegant manipulative experiments that altered sea cucumber population densities and distributions. I fully appreciated the use of historical data to paint the picture of abundant sea cucumbers prior to escalating harvesting efforts.

I've been trying to find weaknesses I can identify but have come up short. However, Figs 3 A, C and D, have three colors to identify the coral outplant experiment. However, some colorblind readers will not be able to distinguish colors among colored points.

We have altered color usage using guidance from an online color palette that avoids red/green colors and specifically is designed to avoid disadvantaging color-blind readers. Hopefully this solves the problem.

This is an important study. Diseases are proliferating among reef corals throughout the tropics but most managers and policy makers feel these are unmanageable. This study suggests a path forward that could curb agents of coral disease transmission.

Finally, this study could have been hopelessly complicated. It includes sediment microbiomes, echinoderm populations, coral, fisheries and disease impacts in geographically broad experimental manipulative studies. It would be very easy to have readers lost in all those weeds. However, I found the paper beautifully written for its clarity and voice. Finally, I appreciate the authors integrating this marine study into the larger picture of terrestrial detritivores. This is often lacking in "marine ecology" studies but it is an important conceptual link that will broaden the pool of readers of this study.

Many thanks for the kind words. We did find this challenging and hope your views of this are universal. We doubt that they will be, but hope springs eternal.

Specific comments:

Line 22:

Farmerfish are territorial damselfish. However, their impacts seem to me to be at distal rather than basal areas of branched acroporids. Is there evidence of the former?

We are confused by this statement. On thicket forming coral like *A. pulchra* (it is very similar to the Caribbean species *Acropora cervicornis* in structure, function, and habitat occupancy if that helps), once the thicket becomes large enough to provide both shelter for damsels and basal coral tissues sufficient for algal gardens, the farmerfishes kill basal portions for their gardens and occupy large thickets in numbers of 100s. Their gardens are always on basal, never on apical, portions. For plate forming *Acropora* species, the reviewer is likely correct because these forms don't provide the same level of 3-D topographic complexity that species like *A. pulchra* and *A. cervicornis* do.

Figure 3. Colorblind readers cannot detect differences in the three-color pattern (e.g., Fig. 3D).

Corrected using a guide designed to allow differentiation by those that are color-blind.

Reviewers' Comments:

Reviewer #1:

Remarks to the Author:

This looks really good. It appears that authors have addressed all major concerns. I think its ready to go.

Reviewer #2:

Remarks to the Author:

The authors did a great job revising the manuscript and responding to all of the reviewer comments. I think this will be an important and highly cited paper!

Two tiny notes:

- Line 92: sentence reads as if the population size is 130km². I suggest revising to "Mo'orea is a high island of 130km² with a human population of about 17,000 people and so might have...."

- Lines 147-148: this sentence is very hard to follow. Not sure if there is a typo or if it just needs to be split into two sentences.

I look forward to seeing the paper in print.

Reviewer #4:

Remarks to the Author:

Overall, I think the revised manuscript addresses most of the concerns expressed by me and by other reviewers. This is a complex story condensed into Nature's format so everything cannot be addressed in detail.

Two points addressed based on my previous review. 1. The figures now are easy for colorblind people to see. 2. Damselfish impacts on cervicorn branched acroporids is not often basal nor apical. However, it is common around the space the damselfish use as their shelter. See Schopmeyer, S.A. and Lirman, D., 2015. Occupation dynamics and impacts of damselfish territoriality on recovering populations of the threatened staghorn coral, *Acropora cervicornis*. PLoS One, 10(11), p.e0141302...